# Technical Aspects of Motor and Language Mapping in Glioma Patients

**DOI:** 10.3390/cancers15072173

**Published:** 2023-04-06

**Authors:** Nadeem N. Al-Adli, Jacob S. Young, Youssef E. Sibih, Mitchel S. Berger

**Affiliations:** 1Department of Neurological Surgery, University of California, San Francisco, CA 94131, USA; nadeem.al-adli@ucsf.edu (N.N.A.-A.); jacob.young@ucsf.edu (J.S.Y.); 2School of Medicine, Texas Christian University, Fort Worth, TX 76109, USA; 3School of Medicine, University of California, San Francisco, CA 94131, USA

**Keywords:** glioma, intraoperative stimulation mapping, motor mapping, language mapping, maximal safe resection, functional brain mapping, awake craniotomy, stimulation techniques

## Abstract

**Simple Summary:**

Intraoperative stimulation mapping is a technique used to identify and preserve functional tissue during the surgical resection of gliomas. This form of functional brain mapping allows neurosurgeons to remove the most tumor tissue possible while minimizing the risk of a neurological deficit after surgery. The data supporting brain mapping and the technical nuances of performing these operations safely are described in this review.

**Abstract:**

Gliomas are infiltrative primary brain tumors that often invade functional cortical and subcortical regions, and they mandate individualized brain mapping strategies to avoid postoperative neurological deficits. It is well known that maximal safe resection significantly improves survival, while postoperative deficits minimize the benefits associated with aggressive resections and diminish patients’ quality of life. Although non-invasive imaging tools serve as useful adjuncts, intraoperative stimulation mapping (ISM) is the gold standard for identifying functional cortical and subcortical regions and minimizing morbidity during these challenging resections. Current mapping methods rely on the use of low-frequency and high-frequency stimulation, delivered with monopolar or bipolar probes either directly to the cortical surface or to the subcortical white matter structures. Stimulation effects can be monitored through patient responses during awake mapping procedures and/or with motor-evoked and somatosensory-evoked potentials in patients who are asleep. Depending on the patient’s preoperative status and tumor location and size, neurosurgeons may choose to employ these mapping methods during awake or asleep craniotomies, both of which have their own benefits and challenges. Regardless of which method is used, the goal of intraoperative stimulation is to identify areas of non-functional tissue that can be safely removed to facilitate an approach trajectory to the equator, or center, of the tumor. Recent technological advances have improved ISM’s utility in identifying subcortical structures and minimized the seizure risk associated with cortical stimulation. In this review, we summarize the salient technical aspects of which neurosurgeons should be aware in order to implement intraoperative stimulation mapping effectively and safely during glioma surgery.

## 1. Introduction

Gliomas are diffuse, infiltrative primary brain tumors, often presenting with seizures or neurological deficits referrable to their location within the brain parenchyma. The current standard of care to improve survival for these patients involves surgical resection followed by chemoradiation for higher-grade malignancies [1]. During tumor resection, the primary goal is to maximize the extent of resection—often performing a supratotal resection (SpTR) of lesional tissue when possible—while also preserving neurological function and patient quality of life [2,3,4,5,6,7]. The benefit of aggressive surgical resections must be balanced with the preservation of neurological function, as neurological deficits, particularly hemiparesis, have been shown to abrogate the procedure’s survival benefits [8,9,10].

Technical and technological advances in the operating room have focused on improving postoperative functional outcomes and intraoperative detection of residual tumor cells to facilitate the goal of maximal safe resection [11]. Unfortunately, these infiltrative tumors are often near functional cortical and subcortical regions, making intraoperative electrical stimulation mapping (ISM) critical to safely remove the lesion [12]. Originally introduced by Penfield [13], the field of brain mapping has advanced to allow for intra-operative identification and preservation of functional tissue during surgery [12,14,15,16,17,18,19,20,21,22,23]. In a large meta-analysis, De Witt Hamer et al. reported ISM to be associated with more extensive resections and fewer late severe neurological deficits, despite more frequently involving tumors located in functional regions [12]. In addition, when compared to asleep resections, awake mapping is associated with fewer neurological deficits, as well as improved overall and progression-free survival [24].

As the arsenal of techniques has expanded and evolved for cortical and subcortical mapping in glioma patients, neurosurgeons need to be aware of the techniques available for safely identifying functional regions for the purpose of maximizing the extent of resection. In this review, we describe the technical aspects of intraoperative mapping, both awake and asleep, for tumor resection.

## 2. Maximizing Extent of Resection Is the Standard of Care

Maximal safe resection is defined as resecting as much tumor-infiltrated tissue as possible to improve survival while minimizing the risk of postoperative neurological deficits and retaining quality of life [2,25,26,27]. As mentioned, SpTR has recently been associated with improved overall survival (OS) for both LGG and GBM [2,28]; however, the survival advantage of these more aggressive resections is lost when patients have a postoperative deficit [8,9,10]. As such, significant efforts have been made to develop pre- and intra-operative methods for maximizing EOR. Furthermore, neurological deficits and poor functional outcomes are associated with the development of medical complications, depression, and an overall poorer quality of life [29,30,31].

Intraoperative stimulation mapping (ISM) remains the gold standard for the identification of functional tissue during surgical resections [32,33]. Preoperatively, it is important to consider several factors, such as tumor localization, the patient’s cognitive and functional status, preoperative neurological deficits, and preoperative anxiety level, when determining the surgical plan. Preoperative tools include functional MRI (fMRI), magnetoencephalography (MEG), diffusion tensor imaging (DTI), and navigated transcranial magnetic stimulation (nTMS) [16,34], which can be used as adjuncts during surgical planning, but they lack the accuracy and specificity to be used in place of intraoperative cortical and subcortical direct electrical stimulation [27,35].

Recent technological advances have improved the reliability of direct cortical and subcortical electrical stimulation, as well as transcranial cortical stimulation [36], which can be used in conjunction with neurophysiological monitoring of motor-evoked potentials (MEPs) and somatosensory-evoked potentials (SSEPs) to provide constant insight into the functional integrity of the corticospinal tract or dorsal columns/medial lemniscus sensory system during tumor resection [33]. Cortical mapping is used to identify functional sites that are vital in the language, motor, somatosensory, and executive/cognitive domains and must be preserved. Non-functional sites revealed as negative sites during mapping can be safely used for the initial corticectomy to approach the tumor [27]. Testing and monitoring of cognitive functions such as language, visual perception, and spatial orientation is dependent on having the patient awake and cooperative during the procedure; therefore, these functions cannot be accurately assessed in asleep craniotomies [27]. While motor mapping may be performed during either awake (AC) or asleep (AS) craniotomies, there is no consensus on the superiority of one technique over the other, and the choice of awake versus asleep mapping often depends on the patient’s symptoms and the tumor’s location and size [27,37,38]. For example, in a matched cohort analysis, Gerritsen et al. reported that awake craniotomies resulted in more extensive resections in the entire cohort, and on subgroup analyses based on cutoffs of 70 years of age, a preoperative National Institutes of Health Stroke Scale (NIHSS) score of 2 and a Karnofsky Performance Scale (KPS) of 90 [24]. In addition, the authors reported OS and PFS benefits in younger patients, as well as those with a NIHSS of 0–1 and a KPS of 90–100. Importantly, two currently recruiting trials aim to assess the safety and efficacy of awake and asleep craniotomies in glioma patients (Table 1), the results of which should aid in corroborating these findings and in surgical decision making.

## 3. Nuances of Intraoperative Motor Mapping Techniques and Measurements

### 3.1. Cortical and Subcortical Motor Mapping

Intraoperative motor mapping is critical for preserving function when resecting tumors near the Rolandic cortex and subcortical corticospinal tract, as postoperative motor deficits have been shown to abolish the survival benefit associated with maximal extents of resection and greatly impair patients’ quality of life [9,10]. Advances in MEP monitoring, neuronavigation, cortical/subcortical mapping, and intraoperative surgical techniques have all contributed to safer resections for tumors near motor regions by minimizing direct damage and/or ischemic injury to these pathways [41,42].

### 3.2. Tractography for the Corticospinal Tract (CST)

In a prospective randomized control trial published in 2007, Wu et al. reported a dramatic improvement in overall survival and better postoperative KPS in patients who underwent resections of tumors involving the pyramidal tracts. This was achieved using DTI fiber tracking of the CST integrated into the neuronavigation, which was more successful compared to standard neuronavigation with structural MRI sequences [43]. While there is no doubt that DTI tractography is helpful for guiding surgeons regarding proximity to vital structures, and is a part of the standard practice for these operations, DTI is not based on physiological parameters; as such, its intraoperative accuracy is subject to numerous limitations. For example, the region-of-interests used as seeds to generate the tractography projects can alter their appearance, and intra-operative brain shift can dramatically impair the navigation/DTI accuracy [23]. As such, DTI alone cannot be used to identify the CST, and instead should be incorporated into the intraoperative decision-making strategy for the pursuit of subcortical mapping.

### 3.3. Motor Evoked Potentials (MEPs)

Intraoperative MEPs are also considered part of the gold standard for supratentorial glioma resections near the primary motor cortex or corticospinal tract. Transcranial MEPs (tcMEPs) utilize a high-voltage electrical stimulus through the scalp/skull to activate the motor cortex and descending pathways to generate an MEP, which can be measured by electrodes on the limbs. Alternatively, MEPs can be obtained by direct cortical stimulation (dcMEPs) after opening the dura by stimulating through a strip electrode that is placed over the primary motor cortex [44]. Following ‘train of 5’ anodal stimulation, a drop in signal amplitude by >50%, with the same stimulus intensity serving as the baseline established at the beginning of the case, or a 20% increase in the stimulus threshold needed to achieve a response compared to the ipsilateral muscle groups, are generally considered to be warning signals for postoperative weakness, and are monitored with the goal of avoiding false negative (i.e., normal MEP signals in patients who ultimately suffer from postoperative weakness) and overly sensitive false positive stimulation results (i.e., drops in MEP amplitude in patients without postoperative motor deficits) [45,46]. MEPs can be obtained every 30 s during the entirety of the tumor resection to monitor the tract integrity, although brain shift as the resection progresses can provide false positive MEP changes that the surgeon can often identify by dynamically returning the cortex to the dural opening with irrigation, manual manipulation, and/or transitioning from tcMEPs to dcMEPs for a more reliable signal.

### 3.4. Awake versus Asleep Motor Mapping

The choice of AC or AS for motor mapping is nuanced, and may be influenced by the patient’s clinical examination and the tumor size/location [47]. There are significant differences in neuroanesthetic regimen, intraoperative neuromonitoring technique, and the complexity of patient tasks between awake and asleep mapping. A recent systematic review reported that both methods are safe in perirolandic tumors [48]; however, AC may offer better EOR and functional outcomes [49]. Alternatively, some studies have argued that patient-level characteristics are the most important for making one method preferable to the other [50]. 

While determining the optimal approach is a multifaceted process, intraoperative mapping techniques in both scenarios have ultimately enabled surgeons to resect tumors that were once considered inoperable within and near the primary motor cortex [51], and as such, have become the gold standard [12]. Nevertheless, there are inherent technical differences between the two methods. For example, in AS, responses are monitored through electrical responses to stimulation and/or passive patient movement, whereas assessment in the awake setting is geared towards impairment during patient-dependent tasks and/or involuntary movement [16,27,50]. Additionally, tcMEPs, which provide the added benefit of motor cortex stimulation without overt craniotomy exposure [46], are modified in awake cases to avoid the pain associated with corkscrew stimulators and subdermal needle EMG electrodes [52]. Rather, in awake craniotomies, stickers can be used for EMG recording, and the ground and reference must be placed close to one another to minimize the amount of current needed to achieve MEPs. Even with these modifications, direct cortical stimulation with a strip electrode on the motor cortex surface with the ground and a reference electrode placed close by is often needed to minimize the current needed to generate MEPs. Finally, given that awake patients do not have bite blocks in place (all asleep patients need to have two carefully placed bite blocks that secures the tongue in the middle of the mouth), the stimulation current must not cause involuntary contraction of the masseter or jaw muscles to avoid tongue lacerations.

Advances in asleep motor mapping, particularly with respect to high-frequency cortical and subcortical stimulation, have resulted in very good functional outcomes following asleep resections of lesions involving the central sulcus and in patients with preoperative weakness [50]. Regarding tumors located within the primary motor cortex, studies have demonstrated that there are no differences between the two with regard to EOR or postoperative morbidity [53,54], while others have suggested that AC is associated with more frequent 100% resections and better postoperative functional status [24,49]. While additional studies are needed to directly compare the two methods, generally speaking, both techniques can be used safely in this high-risk cohort, and outcomes likely depend on the intraoperative techniques used. For example, permanent postoperative deficits have been reported in as few as 2% of AS cases when using adaptive high-frequency monopolar mapping [51], which is preferred in the asleep setting due to the variability of neuromonitoring measurements when the patient is awake [51]. On the other hand, in AC cases, continued resection past the point of failed recovery of an intraoperative deficit is associated with permanent deficits [55]. 

For lesions involving the supplemental motor area (SMA) and motor–praxis network, some groups argue for awake motor mapping, while others advise against AC for lesions in this region, since avoidance of postoperative SMA syndrome is unnecessary given the transient nature of this deficit [50,56]. Still, debate abounds regarding the superior method and the associated risk factors. While the degree of regional resection has been suggested as a risk factor [57], Kumar et al. reported in a small series of SMA region tumors resected using AS, that all tumors were completely resected with intact MEPs. Additionally, despite the development of a case of SMA syndrome, all of their patients recovered completely [58]. Aligned with the hypothesis that these resections can be performed in both settings, Young et al., in their larger cohort of newly-diagnosed SMA region tumors, reported no association between the type of craniotomy (AC vs. AS) and the development of the syndrome [56]. Furthermore, they reported that while larger resection cavities were associated with the development of an SMA syndrome and prolonged recovery, the severity of the symptoms was unaffected [56]. Some studies argue that resection of the frontal aslant tract (FAT) is critical in the development of the syndrome [59], whereas others have suggested that preservation of the FAT is insufficient for prevention [56,60,61]. Alternatively, extensive resection of the posterior SMA region [62,63] and cingulate gyrus are perhaps more important risk factors [56,64]. Taken together, these findings suggest that premature cessation of resection due to insignificant intraoperative deficits may occur in the awake setting [61,65] and without subsequent benefit to the patient. Importantly, during preoperative discussions with their patients, neurosurgeons should highlight the possibility of these temporary deficits when pursuing aggressive resections of tumors in this location, as well as the possibility of these deficits emerging during intraoperative mapping if an awake approach is chosen.

### 3.5. Stimulation Techniques and Nuances 

Motor mapping can be performed using either high- or low-frequency stimulation and either a monopolar or bipolar probe (Table 2). In both paradigms, stimulation aims to identify the lowest intensity at which MEPs are produced (i.e., the cortical or subcortical motor threshold) for accurate boundary identification [52]. Low-frequency bipolar stimulation (biphasic-wave, 1 ms pulse duration, 60 Hz, 4–16 mA) can be used for cortical mapping in order to identify function-free zones for the corticotomy [36]. However, this method only identifies positive sites in subcortical mapping ~40% of the time, giving this technique high specificity, but poor sensitivity [42]. As such, monopolar, high-frequency stimulation is preferred, particularly for subcortical mapping, as it is not only sensitive for functional sites but also provides quantitative estimates of the distance from critical subcortical tracts depending on the intensity of the stimulation used to elicit a response (i.e., 1 mA ≃ 1 mm) [27]. In some instances, a bipolar probe may be used to increase spatial resolution when needed during subcortical mapping [33].

Traditionally, high-frequency monopolar stimulation is administered as a monophasic wave pulse in trains of 5 (0.5 ms pulse duration, 1–4 ms interstimulation interval, 0–20 mA); however, more recently, reduced trains of 1 to 2 [51,66] have demonstrated added utility as well. Rossi et al. directly compared these two protocols and identified tumor subgroups in which shorter trains may offer additional insight. In tumors outside of the primary motor cortex and cortical spinal tracts, those only affecting the cortical spinal tracts, or those originating within the primary motor cortex with normal cortical architecture, ‘train of 2’ stimulation better segregated the anterior and posterior regions of the primary motor cortex—a distinction that may aid in maximizing EOR while preserving long-term function [66]. In addition, only the two trains of stimulation identified function-free zones in primary motor cortex tumors with distorted cortical architecture. The authors identified similar patterns regarding subcortical mapping where two trains of stimulation may have provided added specificity to mapping within the primary motor cortex. In a separate study, Rossi et al. evaluated increased and decreased trains of stimulation and found that in patients with well-controlled seizures and well-defined tumors, maximal safe resection was possible using the standard five trains of stimulation and resecting until reaching a subcortical motor threshold of 3 mA. Alternatively, in those with a complex treatment history, prior seizures, or deficits at presentation, this may need to be adapted by increasing the number of pulses and/or duration of pulses to achieve adequate resection. Lastly, the authors reported that in diffuse tumors, the combination of the five trains and the modified two trains should be used to define functional boundaries at the cortical and subcortical levels [51].

High-frequency monopolar stimulation, first described by Taniguchi et al., has become an emerging tool for cortical mapping and may be delivered in short trains of 3 to 10 (250–500 Hz, 0.5–0.8 ms pulses, 0–20 mA), similarly to subcortical parameters [67,68,69]. However, in contrast to subcortical stimulation, an anodal current should be used, as it more effectively produces corresponding MEPs at lower thresholds [67,68,70]. In comparison to bipolar cortical stimulation, this technique offers equal sensitivity over the primary motor cortex, but in other areas, such as the premotor frontal cortex, bipolar stimulation is superior [68]. The advantages of monopolar cortical stimulation primarily include the lower rate of seizures and the ease of continuing subcortical mapping during resection [71]. Taken together, these techniques are complex and most effective when used in combination with one another to minimize postoperative morbidity [36]. However, neurosurgeons should utilize these methods according to their experience and the available resources.

**Table 2 cancers-15-02173-t002:** Overview of intraoperative motor mapping parameters.

	Cortical		Subcortical		Transcranial
	Monopolar	Bipolar	Monopolar	Bipolar	Scalp Electrodes
Frequency (Hz)	250–500	50–60	250–500	50–60	200–1000
Wave form	Monophasic rectangular	Biphasic square	Monophasic rectangular	Biphasic square	Monophasic
Polarity	Anodal	Alternating	Cathodal	Alternating	Anodal
Intensity	0–20 mA	0–16 mA	0–20 mA	1–6 mA	0–800 V
Duration (ms)	0.5–0.8	1	0.5–0.8	1	0.75
Pulses (Trains)	5–10	60/s	5–9	60/s	3–9
Interstimulus interval (ms)	2–4	16.7	2–4	16.7	1–5
MEP threshold (mA)					
Awake	5–15	2–7			
Asleep	2–7	7–16			
Stimulation amplitudes * (mA) [42,72,73]					
Awake	1–20	2–8	1–20	2–8	
Asleep	1–20	3–16	1–20	3–16	

Abbreviations: Hz, hertz; ms, milliseconds; mA, milliamp. * EcoG should be used with higher stimulation amplitudes to monitor for after-discharge potential, as the risk of a stimulation-induced seizure is higher.

## 4. Sensory Mapping

Somatosensory mapping may be performed in awake or asleep settings, and in the former, may be assessed by patient-reported sensations during electrical stimulation [74,75,76]. Nonetheless, somatosensory evoked potentials (SSEPs) may assist with structural localization and deficit prediction [77]. For the purpose of localizing the central sulcus, phase reversal is one reliable method [78]. Some studies have suggested that SSEPs may additionally be used for monitoring sensorimotor function during glioma resection and predicting neurological deficits [79]. The warning criteria, which were initially defined as a >50% amplitude reduction or >10% propagation of latency from the baseline, have now been adapted to suggest that an obvious, abrupt, and not otherwise explainable visual deviation from pre-change values may be concerning for intraoperative injury [77]. Ultimately, recent literature has reported mediocre predictive statistics associated with these criteria [80], and has failed to demonstrated a significant association with postoperative neurological deficits [81]. Therefore, SSEPs may provide indirect localization of important structures; however, concerning recordings should be interpreted with caution, as they may lead to premature completion of the resection [82]. Moreover, sensory deficits, when reported, have little association with the patient’s functional independence [31], and as such, SSEP monitoring is not necessary given the superiority of MEPs for primary motor cortex identification [33].

## 5. Language Mapping

Awake language mapping is critical for glioma surgery within the dominant hemisphere [15,83,84,85,86]. Previous works have established that language networks are variable, and as such, mapping only when tumors involve specific anatomic regions is inadequate [14,83,87,88] because no structural landmark on preoperative MRI can precisely predict functional tissue [84]. Combined with the possibility of functional tissue being present within the tumor region [89,90] and tumor-induced reorganization [91,92,93,94], these factors complicate language localization [95]. While ISM is the gold standard, intraoperative MRI (iMRI), 5-aminolevulinic acid (5-ALA), and intraoperative ultrasound (iUS) are notable adjuncts that can be used in the operating room to increase the extent of resection [96,97,98].

### 5.1. Patient Selection and Preoperative Assessment

Although no guidelines on patient selection for awake surgery exist, some contraindications include uncontrolled coughing, severe dysphagia, and greater than 33% naming errors despite dexamethasone and mannitol treatment [99]. Some commonly cited relative contraindications include significant diuretic- and steroid-resistant mass effect, obesity (BMI > 30), psychiatric and/or emotional instability, under 10 years of age, intraoperative seizures, current smoker, intraoperative nausea, reoperation, and significant preoperative functional impairment; however, specific solutions for each relative contraindication have been described to safely perform awake procedures in patients with these comorbidities or conditions [99,100].

Preoperative evaluation includes anatomic imaging, diffusion tensor imaging tractography, functional connectivity maps with magnetic source imaging (MSI) and magnetoencephalography (MEG), neurolinguistic testing, patient counseling, and, in some cases, a neuropsychological evaluation. Additional functional imaging techniques may be used for preoperative language mapping, such as fMRI and nTMS, although these noninvasive imaging modalities are not specific enough to determine the location of language function beyond hemispheric dominance [17,101]. 

### 5.2. Anesthetic Considerations

While a dedicated neuroanesthesia team is essential to providing the optimal care for patients, various neuroanesthetic regimens can be used during an awake craniotomy [102]. A recent meta-analysis reported that the commonly used asleep–awake–asleep (AAA) and monitored anesthesia care (MAC) techniques are equally safe [103]. Propofol-remifentanil and/or dexmedetomidine may be used for the AAA approach [99,104,105], while dexmedetomidine is typically the only agent used in MAC cases. While a nasal cannula is used for supplemental oxygen in all cases, a laryngeal mask airway or nasal trumpet should be available and used when needed [99,103]. Generous local analgesia is essential for Mayfield placement, and a scalp block can be useful for pain relief prior to skin incision [100].

### 5.3. Current Technique

A focused exposure begins with a tailored craniotomy site over the lesion and any adjacent structures that may require mapping. Dural opening is typically more challenging in reoperation cases due to dural scars and adhesions. During the dural opening process, lidocaine may be used to provide a dural block, particularly near the middle cranial fossa floor, if the patient is experiencing discomfort. Sedation is significantly reduced or stopped all together prior to opening the dura, and once adequate cortical exposure is achieved, anesthesia should have propofol in line and iced Ringer’s solution should be available for seizure control if needed. In both techniques, a short assessment of the patient’s wakefulness is performed prior to cortical mapping and linguistic testing. Regarding task selection, these vary widely between institutions and there is currently no agreement on the optimal test to use. Picture naming, text reading and writing, sentence completion, syntax, auditory naming, and spelling are some of the most common assessments performed [100]. While language assessment protocols exist to standardize intra-operative task selection [106], they are not widely used, and ultimately, care should be taken to avoid tests with poor sensitivity or specificity during baseline testing [106,107].

Various techniques for cortical and subcortical mapping have been reported using varied parameters [108,109]. Low-frequency bipolar stimulation (60 Hz, 1.25 ms biphasic square waves in 4 s trains) generated across 1 mm electrodes separated by 5 mm is traditionally used. However, some studies have reported using high-frequency monopolar stimulation (HFMS) for language mapping with comparable results when utilizing high-frequency trains at a repetition rate of 3 Hz [110,111]. Compared to low-frequency bipolar stimulation (LFBS) for motor mapping, HFMS is known to be more efficacious and less likely to induce intraoperative seizures [111], which makes its potential implementation for language mapping intriguing. However, in the few studies that have directly compared the two methods, seizures occurred in 7–11% of patients [110,111], which suggests that additional data are still needed.

Nonetheless, standard cortical mapping typically begins at a 2 mA stimulus, which can be increased until somatosensory or motor function is identified or, in the case of language, until after-discharge (AD) or epileptiform activity are noted on electrocorticography (ECoG) by an epileptologist. In the case of language mapping, classically, the AD-induced intensity is reduced by 1 mA and then used for the remainder of the language mapping process, which usually ranges from 3 to 4 mA, to avoid false positive results from AD-induced errors and minimize the risk of seizures [112]. If motor or somatosensory sites are not exposed, a 4-contact strip electrode can be advanced subdurally to establish positive somatosensory/motor sites.

Cortical testing sites, separated by 1 cm, are non-sequentially tested 3 times each for 3 to 4 s, with a 4 to 10 s inter-task interval. If patients are fatigued or struggling with the testing, the inter-task interval can be prolonged to give the patients more recovery time between tests. A site is considered ‘positive’ when it produces either speech arrest without a simultaneous motor response, anomia, or alexia in two of the three attempts [104,113]. A trained neuropsychologist engages with the patient while coordinating with the neurosurgeon during mapping to identify positive and negative sites. These are recorded along with the stimulation parameters and marked using numbered indicators. Cortical dissection using an ultrasonic aspirator proceeds through ‘function-free’ corridors, while a 1 cm margin should be preserved around ‘positive’ sites [99,114]. Importantly, we found that this method of negative mapping has an exceedingly low false-negative rate [99], and as such, if cortical mapping reveals no ‘positive’ sites, greater exposure to find a ‘positive’ site is not necessary [15]. Subcortical mapping is performed in a similar fashion, but is focused on nearby areas with presumed language function, for which preoperative tractography superimposed within the intraoperative neuronavigational space can be useful [115,116]. Critical subcortical tracts involved in language include the arcuate fasciculus (AF), superior longitudinal fasciculus (SLF), inferior longitudinal fasciculus (ILF), inferior fronto-occipital fasciculus (IFOF), uncinate fasciculus (UF), and subcallosal fasciculus (SF) [117]. In addition, each subcortical tract and their associated pathways are responsible for highly specific functions that are individually and collectively important for language and conflictive function [118]. As such, an individualized approach is taken when choosing tasks and stimulation sites which is tailored towards the characteristics of both the patient and the tumor [116,118].

## 6. Executive Function—Beyond Language and Sensorimotor

Executive function (EF) describes the way in which one can coordinate and control higher-order behaviors, social abilities, and cognitive tasks. Despite advances in neuroimaging and brain mapping that have resulted in improvements in functional outcomes and EOR, patients may still develop an array of cognitive deficits that impact their quality of life, ability to return to work, and capacity to complete activities of daily living [119]. Although a recovery of cognitive function to the preoperative level is possible, much of the literature lacks robust discussion of EF [120]. In addition, a complete understanding of the cortical and subcortical networks involved in EF has not yet been achieved; however, functional imaging has revealed that these networks are primarily located in the frontocorticostriatal region of the brain [25]. The incomplete understanding of these networks and their importance to postoperative patient performance status and quality of life makes intraoperative mapping a relatively complex challenge, and likely explains the limited research on the subject [121]. In addition, EF testing is intricate and time-consuming, which may reduce patient cooperation during the awake portions of their surgery [25,118,122].

Despite these challenges, studies have begun to evaluate the feasibility of EF mapping, with a primary focus on LGG patients due to their potential for deeper subcortical infiltration and longitudinal impacts on cognition [4,123] (Table 3). Accordingly, the frontoparietal and the frontal cortico-subcortical networks along with the FAT have been shown to have roles in executive function, and, when disrupted during LGG resection, may be implicated in EF deficits [124]. Wager et al. were the first to report the operative feasibility of the Stroop Test, which is a well-established tool for evaluating executive function during cortical mapping [125]. Puglisi et al. added to these results by demonstrating the efficacy of a simplified version, the “intraoperative version of the Stroop task” (iST), during subcortical mapping [126]. Their study revealed that iST-positive subcortical sites were correlated with executive function and, when spared, patients experienced minimal deficits at their 3-month follow-ups. Importantly, the implementation of this task did not affect the extent of resection [126]. Erez et al., in their novel implementation of ECoG monitoring, demonstrated the potential of resection to support and guide direct electrical stimulation in order to identify the functional regions of the cortex which are involved in EF [127].

Taken together, considering the diverse array of EF-related behaviors and the limited time frame allotted for functional mapping during an awake craniotomy, it would be extremely difficult to assess all aspects of this domain. Therefore, it would be most appropriate to perform comprehensive preoperative neuropsychological batteries to identify the most patient-centered, clinically relevant functions in order to potentially assess intraoperatively. Ultimately, however, larger, prospective studies assessing EF with an intent to provide clinically relevant improvement are needed before the wide implementation of such techniques is possible.

## 7. Managing Expected and Unexpected Intraoperative Events

### 7.1. Intraoperative Seizures

A risk associated with direct cortical stimulation during intraoperative mapping is the development of stimulation-induced seizures [131], occurring in 2.5 to 54% of awake craniotomies [132]. These may complicate the operation and are the leading cause of aborted awake operations [133], although their incidence is relatively low [134]. Intraoperative ECoG analyses have previously demonstrated that intraoperative seizures and after-discharges can be avoided by limiting the charges transferred per second and the total number and duration of stimulations [132]. A practical approach for management involves the application of cold Ringer’s solution to the cortex until cessation [131]. Rarely, recurrent seizures may require propofol and/or laryngeal mask airway intubation for airway protection [135]. Neither intraoperative nor after-discharge seizures have a significant effect on the presence of postoperative neurological deficits, length of hospital stay, or perioperative seizure activity [136].

### 7.2. Changes in Neuromonitoring or Task Performance

Currently, there are no set criteria for classifying a significant intraoperative change in the setting of MEPs. The American Society of Neurophysiological Monitoring has published a position statement that a marked reduction in the amplitude of the evoked response, acute threshold elevation, and signal disappearance are indicators of potential motor injury [137]. As previously mentioned, similar warning criteria exist for SSEPs, but with limited reported utility. Nonetheless, the monitoring of MEPs have demonstrated utility in predicting and helping to prevent motor tract injury [138,139], particularly related to ischemia. In the event of MEP deterioration or loss, in most instances, the resection should be stopped and the field evaluated for any potentially obvious causes. Subsequent actions may include irrigation, filling the cavity with fluid to reduce brain shift, papavarine to treat vasospasm of small lenticulostriate vessels, relaxing any fixed retractors, and/or complete cessation of resection in the associated region [138].

In addition to neuromonitoring and preoperative deficits, intraoperative performance on selected tasks and/or positive mapping sites may also be associated with postoperative deficits [85,140]. A recent systematic review reported that intra-operative anomia and production errors were significantly predictive of postoperative language deficits in the acute phase (1 to 10 days), and when combined with a preoperative deficit, the probability further increased [140]. Importantly, these factors were not associated with postoperative deficits at 3 to 8 months. Similarly, identifying positive subcortical sites during motor mapping and preoperative deficits have been reported as independent risk factors for transient or permanent postoperative motor deficits [85]. When both factors were present, there was a significant increase in the odds of a transient deficit; however, for permanent deficits, only the presence of a positive bipolar subcortical site was a risk factor. As such, neurosurgeons should pay particular attention to these specific intraoperative findings during resection and utilize them to guide further surgical decision-making, as well as during postoperative patient counseling regarding expectations.

### 7.3. Avoiding Intra-Operative Awake Craniotomy Failures

Awake craniotomies require intensive preparation and precise timing of the patient awakening’s to avoid intraoperative complications and reduced patient cooperability [27]. Failure rates have been reported to be as high as 6.4%, and are associated with poor preoperative patient selection and adverse effects from intraoperative medications [134]. Although studies have reported that emergency intubations rarely occur [141,142,143,144], seizures and respiratory complications are most frequently the cause [134]. Alternatively, communication-related failures occur more frequently, and are associated with preoperative deficits and functional status [134]. As such, preoperative evaluations in order to predict intraoperative difficulties with an AC are currently under development [145].

Other intraoperative major events, such as respiratory or hemodynamic events requiring intervention, have been associated with remifentanil infusion, increased duration of tumor resection following cortical mapping, and a history of asthma [146]. Regarding the neuroanesthetic technique, Eseonu et al. reported shorter mean operative times in MAC versus AAA awake craniotomies (283.5 min vs. 313.3 min; *p* = 0.038); however, there were no differences in mean length of stay or rate of conversion to general anesthesia between the two groups [147]. Finally, in a large cohort study of 611 patients undergoing awake craniotomy over a 27-year period, Hervey-Jumper et al. reported that neither tumor location, ASA classification, tumor pathology, seizure history, Mallampati score, smoking status, nor BMI impacted the safety or efficacy of awake operations [99]. Taken together, awake operations are generally safe when performed by an experienced neurosurgical team and with proper preoperative evaluation and patient counseling.

## 8. Conclusions

Gliomas are a highly prevalent cause of major disability and mortality across the globe. During the surgical management of this disease, neurosurgeons should aim to resect the maximal amount of tumor-infiltrated tissue while preserving motor, sensory, language, and cognitive function to provide patients with the best quality of life. A deep understanding of the technical, anatomical, and functional nuances is needed to safely resect these infiltrative tumors. Intraoperative stimulation mapping is a safe and effective method for achieving these goals; however, it requires a multifaceted and patient-centered approach during surgical decision-making. Finally, regardless of the techniques or additional adjuncts implemented by brain tumor neurosurgeons, emphasis should always be placed on feasibility and safety, all while considering the patient’s goals for their care. 

## Figures and Tables

**Table 1 cancers-15-02173-t001:** Ongoing clinical trials for glioma surgery.

Study Name (NCT)	Interventions	Primary Endpoint	Secondary Endpoint	Est. Enrollment	Start Date	Est. End Date
PROGRAM (NCT04708171) [39]	Awake mapping Asleep mapping Asleep no mapping	NIHSS, EOR	OS, PFS, Onco-functional outcome, SAE, RTV, MRC motor	453	1 January 2022	1 October 2026
SAFE (NCT03861299) [40]	Awake craniotomy Asleep craniotomy	NIHSS, EOR	EQ-5D, EORTC-QLQ-BN20/C30, OS, PFS, SAE	246	1 April 2019	1 April 2024

Abbreviations: NIHSS, National Institutes of Health Stroke Scale; EOR, extent of resection; OS, overall survival; PFS, progression-free survival; SAE, serious adverse events; RTV, residual tumor volume.

**Table 3 cancers-15-02173-t003:** Intraoperative tasks for assessing executive function.

Task	Function	Result
ST/iST [125,126]	Selective attention and inhibition	Feasible and associated with improved deficits at 3 months
WAIS-III-PA [128]	Social cognition	Feasible and associated with maintenance of baseline performance at 3 months
mJFE [129]	Basic emotion	Positive sites preserved, postoperative decline in function, 3-month improvement
Facial expression pictures [130]	Emotional recognition	No postoperative deficits when positive sites were preserved

Abbreviations: ST, Stroop test, iST, intraoperative Stroop test; WAIS-III-PA, Wechsler Adult Intelligent Scale, 3rd edition—picture association; mJFE, modified Japanese facial expressions of basic emotions test.

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
