# Peer review of "Technical Aspects of Motor and Language Mapping in Glioma Patients"

_cancers, 2023, doi:10.3390/cancers15072173_

Round 1

Reviewer 1 Report

The authors provide a review on mapping techniques during glioma resection. The team this work comes from is for sure leading in the field, which is being reflected in the excellent quality of the manuscript. I only have small comments and believe this adds value to the literature for clinicians not being too familiar with current mapping approaches.

Title:
As the authors also touch on somatosensory mapping, the title might be a little too narrow. I would propose to re-consider.

Simple summary:
"This Intraoperative stimulation mapping..." - please remove either "this" or replace with "the".

Abstract:
"...removed to approach the glioma's equator." - this reads weird, please rephrase.

2. Maximizing extent of resection is the standard of care 
- reference #39 refers to the PROGRAM study in the text, while data from the GLIOMAP study are being cited. Please resolve.
- the GLIOMAP study should be cited in the text and the superiority of awake vs. asleep resection in the presented glioblastoma cohort should be explicitly outlined.

3.3. Motor evoked potentials
- how much of an increase in the threshold needed to achieve a response compared to the ipsilateral muscle groups is considered a warning sign. Please quantify based upon literature.

3.4. Asleep versus awake motor mapping
- "while others have suggested that AC are associated with more frequent 100% resection.." - please also cite #39 (Gerritsen et al. in Lancet Oncol, 2022) here.
- in the context of SMA syndrome, I agree that premature cessation due to clinically irrelevant warning signs might be a problem. However, I believe it is important to point out that peculiar patient guidance is needed to inform patients that short-term deficits may occur when aggressive resection is being provided. Please point that out.

3.5. Stimulation techniques
- "...segregated the anterior and posterior regions." - which regions? from what?

Additional:
the abbreviation of "ECoG"  needs introduction in chapter 6 and not in chapter 7.

Reviewer 2 Report

I would like to thank you for the opportunity to review this article. 

It is an easy-to-understand review that well summarizes the state of the art of intraoperative mapping.

Reviewer 3 Report

Dear Authors,

thank you very much for your work. It is a compact and very nice summary of actual monitoring techniques in glioma surgery for both awake and asleep surgeries. The manuscript is good structured including clearly all aspects from anesthesia, through surgery , technical aspects up to complications. I would like to have three questions which are not mandatory from my point of view. In regard to resection of infratentorial tumor and brain stem involving tumor. Sometimes, due to transcranial MEP-measurement and large CSF lost, early MEP instability may lead to early resection stop. Sometimes might be this monitoring slightly misleading and especially in younger patient leave more tumor behind which can again influence negatively the outcome. Do you have some comments on that? Considering awake surgeries, how do you see the role of intraoperative imaging techniques US/5-ALA or MRI in relation to different monitoring strategies? Are there special criteria for selection of one or another battery of intraoperative tasks assessing the functional outcome?
